# Propagation of a De Novo Gene under Natural Selection: Antifreeze Glycoprotein Genes and Their Evolutionary History in Codfishes

**DOI:** 10.3390/genes12111777

**Published:** 2021-11-09

**Authors:** Xuan Zhuang, C.-H. Christina Cheng

**Affiliations:** 1Department of Biological Sciences, University of Arkansas, Fayetteville, AR 72701, USA; 2Department of Evolution, Ecology, and Behavior, University of Illinois, Urbana-Champaign, IL 61801, USA

**Keywords:** de novo gene, new gene evolution, repetitive protein, gene family expansion, evolutionary process, molecular mechanism, gadid, antifreeze glycoprotein

## Abstract

The de novo birth of functional genes from non-coding DNA as an important contributor to new gene formation is increasingly supported by evidence from diverse eukaryotic lineages. However, many uncertainties remain, including how the incipient de novo genes would continue to evolve and the molecular mechanisms underlying their evolutionary trajectory. Here we address these questions by investigating evolutionary history of the de novo antifreeze glycoprotein (AFGP) gene and gene family in gadid (codfish) lineages. We examined AFGP phenotype on a phylogenetic framework encompassing a broad sampling of gadids from freezing and non-freezing habitats. In three select species representing different AFGP-bearing clades, we analyzed all AFGP gene family members and the broader scale *AFGP* genomic regions in detail. Codon usage analyses suggest that motif duplication produced the intragenic AFGP tripeptide coding repeats, and rapid sequence divergence post-duplication stabilized the recombination-prone long repetitive coding region. Genomic loci analyses support *AFGP* originated once from a single ancestral genomic origin, and shed light on how the de novo gene proliferated into a gene family. Results also show the processes of gene duplication and gene loss are distinctive in separate clades, and both genotype and phenotype are commensurate with differential local selective pressures.

## 1. Introduction

Evolutionary adaptation relies on the origination of new genetic elements that could fuel evolutionary innovation. A type of new gene—de novo gene, which arose from DNA sequences that were ancestrally non-genic has gained increasing recognition as widespread, paving the way for a potential paradigm shift in our understanding of how new genes evolve. Emergence of a de novo gene from a non-protein coding sequence, previously deemed as a rare event [1,2], has been proposed to be a dominant mechanism of novel gene origination in recent analyses [3]. Occurrence of de novo genes could be quite readily inferred by comparative genomic and transcriptomic sequence analyses, but elucidating the molecular mechanism of their evolutionary process, the functional properties of the proteins they encode, and the adaptive fitness they confer are far more challenging [4,5,6]. An immediate, difficult conundrum is the usual lack of an a priori clue of the de novo gene’s function, hampering experimental hypothesis testing. The expected rapid evolution of de novo gene sequences could erase their similarity to the associated ancestral non-coding DNA. Most of the formative de novo genes are thought to be unable to gain a function before their gene-like properties decay [7,8], unless the new random peptides are beneficial enough to achieve a significant fitness shift, allowing natural selection to act on.

A fortunate exception to these uncertainties is the diverse novel antifreeze proteins that evolved in various polar teleost fish lineages. The first antifreeze—the antifreeze glycoproteins (AFGPs) of Antarctic notothenioid fishes—was discovered 50 years ago [9], and was followed in rapid succession by discoveries of near-identical AFGPs in unrelated northern cods, and other structurally distinct types of antifreeze peptides (AFPs) in divergent fish taxa [10]. The protein function and organismal fitness consequence were elucidated prior to (some concurrent with) the advent of molecular technologies to address questions of evolutionary origins and processes [11]. Regardless of structural differences, AF(G)Ps endow the fish with the same life-saving adaptation of freezing avoidance, allowing them to exploit icy, frigid polar and subpolar waters otherwise perilous to unprotected species. The environmental selective force for antifreeze evolution stems from a simple constraint pertaining to a colligative property of liquids. Marine teleosts are hyposmotic to seawater, with typical colligative freezing points (FPs) of −0.7 °C to −1.0 °C, higher than the f.p. of seawater (−1.9 °C). In seawater colder than a fish’s colligative f.p. and with ice present, the fish cannot avoid inoculative freezing of its body fluids by environmental ice. This mismatch between colligative FPs consigns species confined to icy, freezing waters by geography or life history to inescapable death from freezing. AF(G)Ps protect the fish by their ability to recognize ice crystals that enter into the body, bind to them and inhibit ice growth, thereby preserving blood and body fluids in the liquid state [9,10].

The evolution of AF(G)Ps in polar and subpolar teleosts must have been compelled by this strong life-or-death selective pressure stemming from oceanic glaciation that dated to relatively recent geological times, about mid-Miocene (~15 mya) in the Southern Ocean and late Pliocene (~3 mya) in the Arctic seas. Thus, this group of young new protein genes, minted under a clear causative environmental driving force, with established adaptive function and fitness benefit, diverse structural types indicating distinct origins, and tractable ancestral sequences due to their evolutionary recency, comprise an exemplar system for understanding the breadth of origins and mechanisms by which novel genes could arise.

The elucidation of how fish AF(G)Ps arose has indeed broadened our view of evolutionary ingenuity ([11], and references therein). AFGPs evolved convergently from different genomic origins in two unrelated groups of fishes inhabiting opposite poles [12]—the Antarctic notothenioids that are modern perciform fishes, and the Arctic and northern codfishes of the much older gadiform lineage [13]. The protein sequences of AFGPs from these two divergent taxa are near indistinguishable. In both, they occur as a family of size isoforms, all composed of repeats of a conserved glycotripeptide unit, (Thr-Ala/Pro-Ala)_n_. The glycosylation is a disaccharide of galactosyl-N-acetylgalactosamine O-linked to each Thr residue, and the number (n) of repeats range from four to ≥55 depending on species [9,14,15]. As such, AFGPs of the unrelated notothenioid and gadid fishes represent a rare case of protein sequence convergence at the whole protein level.

The Antarctic notothenioid AFGP arose from a functionally unrelated trypsinogen, but the repetitive coding sequence for the tripeptide repeats responsible for the ice binding function originated de novo, from duplications of a partly non-coding 9-nt sequence straddling an intron-exon junction of the trypsinogen ancestor, followed by shedding the unused trypsinogen coding segments [12,16]. Thus, notothenioid *AFGP* evolution was generally recognized as the first clear example that a new functional coding sequence (for a crucial life-saving adaptation in this case) could arise de novo [17]. Underscoring the power of a key innovation driven by climatic change, *AFGP* evolution in the ancestral Antarctic notothenioid enabled its remarkable diversification and adaptive radiation into otherwise uninhabitable icy niches to become the predominant fish group in the Southern Ocean today [18].

The genesis of gadid *AFGP* proved to be even more interesting than notothenioid *AFGP*; it was fully de novo, from entirely non-protein coding DNA [19]. Our prior study identified the ancestral site and intermediate sequences, and deduced the series of molecular mechanisms involving point mutation, tandem duplication, frameshift mutation, and genomic rearrangement that would produce the necessary genic components for transcription, translation, and protein secretion in the de novo formation of the new AFGP gene. In this study, we further investigate how the de novo gadid AFGP gene developed the essential cleavage sites in its polyprotein precursor for producing different protein size isoforms, and how the new gene became stabilized and propagated into a gene family. We also assess whether the AFGP genes in different AFGP-bearing species evolved through distinct trajectories, and how the gene family expansion related to differential local selective pressures. To address these questions we determined the AFGP activity of a broad sample of codfishes, characterized the AFGP abundance and isoforms of AFGP-bearing species, reconstructed a gadid phylogeny of sampled species, and mapped the AFGP phenotype on the phylogenetic framework. We selected three representative species from the three different AFGP-bearing subclades, characterized their AFGP genes and *AFGP* loci expansion to deduce genotypic and phenotypic evolution of AFGP in these lineages.

## 2. Materials and Methods

### 2.1. Specimen and Sample Collection

Species, collection season and locality information are listed in Appendix A. Eighteen gadid species were sampled, including 11 of 12 genera of subfamily Gadinae, two of six genera of subfamily Gaidropsarinae, and two of three genera of subfamily Lotinae. Tissues were dissected from MS-222 anaesthetized fish on ice, preserved in 90% ethanol and kept at −20 °C, or frozen and stored at −80 °C until used for DNA extraction. Blood serum samples were obtained for testing antifreeze activity. Blood was withdrawn from the caudal blood vessel of MS-222 anaesthetized fish using an appropriate size needle and syringe and allowed to coagulate at 4 °C. After centrifugation to pellet the clot, the serum was transferred to a new tube and kept frozen at −80 °C until testing. The handling and sampling of fish followed University of Illinois IACUC approved protocol 09138.

### 2.2. PCR Amplification of mt COI and Sequencing

Genomic DNA was extracted from liver or spleen tissue, using standard proteinase K digestion followed by phenol–chloroform extraction and ethanol precipitation methods.

To construct a gadid phylogenetic tree, we amplified and sequenced full-length mitochondrial COI genes from the 18 gadid species. Primer pairs were designed from sequence alignments of the tRNA genes flanking COI in teleost fishes. The primers sequences are:

COI-Forward: 5′-GCCTCGATCCTACAAACTCTTAGTTAACAGC-3′,

COI-Reverse: 5′-GGCTTGAAACCAGYTYATGGGGGTTC-3′.

PCRs were carried out in Eppendorf Mastercycler, with reaction volumes of 50 μL containing ~1 μg of genomic DNA, 0.2 mM dNTPs, 1 μM each primer, 2 mM MgCl_2_, 5 μL 10 × reaction buffer, and 2 U Taq polymerase. The following cycling profile was used: 95 °C initial denaturation for 3 min; 35 cycles of 95 °C denaturation for 45 s, 54 °C annealing for 45 s, and 72 °C elongation for 100 s; and a final extension at 72 °C for 7 min. PCR products were either treated with SAP/EXOI (New England Biolabs, Ipswich, MA, USA) and directly sequenced with the PCR primers, or cloned into the pGEM^®^-T Easy vector (Promega, Madison, WI, USA) and then sequenced. Sequencing reactions utilized BigDye v.3.1 chemistry (Applied Biosystems, Waltham, MA, USA), and were read on an ABI3730xl automated sequencer at University of Illinois Biotechnology Center (Urbana, IL, USA). Partial COI sequence reads were assembled using ChromasPro v.1.42 (Technelysium, Brisbane, Qld, Australia) with manual editing. The COI sequences have been deposited to NCBI, and the accession numbers are listed in Appendix A.

### 2.3. Gadid Phylogenetic Analyses

Two representatives of the gadoid family Macrouridae, *Squalogadus modificatus* (subfamily Macrouroidinae) and *Trachyrincus murrayi* (subfamily Trachyrincinae), were used as outgroups. The sequences of these species were obtained from GenBank (Accession number: NC_008223 and NC_008224).

Combined with the ND2 gene sequences from our previous study [19], the concatenated sequences of COI and ND2 genes of 20 species were aligned with codon constraint using MUSCLE v.3.8.31 [20]. Alignment of each gene was also made to test for their respective best evolution models of nucleotide substitution using ModelTest-NG v.0.1.6 [21]. Likelihood ratio tests were conducted to compare different models and then the best model was selected based on corrected Akaike information criterion (AICc) and Bayesian information criterion (BIC). The model HKY+I+G was selected for COI gene, and TrN+I+G was selected for *ND2* gene. Therefore, the sequence partitions of each gene in the concatenated sequence alignment were analyzed using their gene-specific models in Bayesian and Maximum likelihood phylogenetic analyses.

Partitioned Bayesian phylogenetic analysis was performed with MrBayes v.3.2.6 [22,23]. Three independent analyses were run with Markov chain Monte Carlo (MCMC) sampling with four chains. Tree and parameters were sampled every 100 generations over a total of 100 million generations. Then convergence to the stationary distribution of MCMC was evaluated using standard deviation of clade frequencies and potential scale reduction factor [23], with the first 25% of the sampled trees discarded as burn-in. Finally, the 50% majority consensus tree with Bayesian posterior probabilities for nodes was computed. Partitioned maximum likelihood (ML) phylogenetic analysis was performed with IQ-TREE v.2.1.3 [24,25]. Node supports were evaluated with 1000 replications of standard nonparametric bootstrap. Maximum parsimony (MP) analysis was conducted with PAUP v.5.0 [26] using the heuristic search option, with 1000 random stepwise addition sequence replicates generated under the tree bisection reconnection (TBR) branch swapping algorithm. Support for the internal branches was evaluated via 1000 bootstrap replications. Neighbor-joining (NJ) analysis was conducted with MEGA X program [27] using the maximum composite likelihood method with 1000 bootstrap replications. All codon positions, both transitions and transversions were considered as informative sites. Gamma distribution was selected as rates among sites. Insertion and deletion sites were deleted pairwise.

### 2.4. AFGP Trait Characterization and Mapping to Phylogeny

We measured antifreeze activity of blood serum samples of all gadid species except *Lota lota* (serum sample was unavailable) used in the study. We used a Clifton nanoliter freezing point osmometer (Clifton Technical Physics) equipped with a cryoscope, which permits direct observation of temperature-dependent ice crystal growth morphology and behavior in the sample [28]. Briefly, submicroliter droplet of a serum sample was suspended in immersion oil in the sample holder housed in a Peltier thermoelectric unit. The sample was first snap frozen at −40 °C, and then gradually warmed to melt the ice back to a temperature at which a minute single ice crystal (~5–10 µm wide) stably remained. This temperature is the equilibrium (colligative) melting/freezing point (eqMFP) of the sample. The single ice crystal becomes hexagonally faceted if AFGPs are present even at very low levels, but remains as a rounded disc if antifreeze is absent. In samples with substantial AFGPs, ice growth is strongly inhibited as the temperature is gradually lowered, and the ice crystal remains small and assumes a hexagonal bipyramid morphology. Upon cooling to the non-equilibrium melting/freezing point (non-eqFP), burst growth in form of ice spicules from the hexagonal bipyramid occurs. The numerical difference between the eqMFP and non-eqFP measures the magnitude (°C) of antifreeze activity, also termed thermal hysteresis. Serum thermal hysteresis of high Arctic gadids reach can reach well over 1 °C. Samples with little (or ineffective) antifreeze cannot retard the growth of the hexagonal ice crystal disc when cooled even slightly below eqMFP, thus there is insignificant or no measurable thermal hysteresis. Serum samples without antifreeze do not facet the test ice crystal nor produce thermal hysteresis.

For gadids with substantial thermal hysteresis (≥0.4 °C), we purified the AFGPs from their blood serum to characterize and compare AFGP isoform composition between species, following published protocols [29]. Briefly, half to one milliliter of serum was reacted with equal volume of 5% TCA (trichloroacetic acid), which precipitates all serum proteins except the TCA-resistant AFGPs. After centrifugation, the supernatant containing soluble AFGPs was transferred into a Spectrpor3 (Spectrum) dialysis tubing (MWCO 3000 Da), and dialyzed exhaustively in distilled water at 4 °C. Dialyzed AFGPs were lyophilized to dryness and weighed. The native circulating concentration (mg/mL) of AFGPs in blood plasma was estimated by dividing the dry AFGP mass by the known serum volume they were purified from. To resolve and visualize AFGP isoforms, purified AFGPs were fluorescently labeled and electrophoresed on a native, discontinuous tris-borate polyacrylamide gel following published protocols [30]. AFGPs do not bind SDS, nor any of the typical protein stains, necessitating fluorescent labeling and borate buffer to impart charge for mobility. Briefly, purified AFGPs were dissolved in sample buffer (450 mM boric acid, 30% glycerol, pH 8.6) to 50 ug/uL, and a volume containing 300 to 400 ug were N-labeled with 1 uL fluorescamine (4 mg/mL in acetone) (Sigma-Aldrich, Burlington, MA, USA). The labeled sample was lectrophoresed on a discontinuous polyacrylamide gel of 4% stacker gel (pH 6.8) and 10–20% gradient resolving gel (pH 8.8) with a tris-borate running buffer (2.5 mM Tris, 200 mM glycine, 1 mM boric acid) at 25 mA constant current. The gel with the resolved fluorescent AFGP isoforms was then imaged using a Kodak EDAS 1D gel documentation system.

The AFGP phenotype is mapped to each species in the gadid phylogenetic tree we constructed. Symbolic designations of phenotype status, namely minus (−) or plus (+) with gradations was based on thermal hysteresis measurements, temperature-dependent ice crystal growth morphology and behavior, and yields of AFGPs and integrity of the AFGP isoforms purified from the gadid species.

### 2.5. Characterization and Comparison of AFGP Loci and Neighboring Genomic Regions

We analyzed the *AFGP* loci and flanking genomic regions of three cold-water gadid species with AFGP phenotype, *Boreogadus saida, Gadus morhua*, *Microgadus tomcod*. The genomic sequences encompassing the large *AFGP* loci (GenBank Accession MK011258-MK011262) were obtained from sequencing of BAC (Bacterial Artificial Chromosome) clones identified from BAC libraries as described in our prior studies [19,31]. Pairwise alignments and synteny analysis were performed using the genomic sequence alignment tool wgVISTA [32], the local alignment tool BLASTN, and the global alignment tool MUSCLE v.3.8.31 [20]. We annotated gene contents in the reconstructed *AFGP* loci and homologous genomic regions using BLAST tools, and gene prediction tool FGENESH V2.6 (http://www.softberry.com/, accessed on 18 July 2021). Known repetitive and transposable elements (TE) were identified and masked using RepeatMasker Version 4.0.1 [33].

### 2.6. AFGP Gene Feature and Phylogenetic Analyses

We extracted the sequences of AFGP genes and pseudogenes from the assemblies and characterized their gene features, including length, signal peptide, types of polyprotein cleavage sites and their frequencies, exon-intron boundaries, amino acid composition, and C-terminus non-tripeptide residues.

We constructed the AFGP gene tree to identify orthology between AFGP gene copies among the three gadids and determine the evolutionary relationship of the gene family members within each species. Because the repetitive coding regions are impossible to be aligned with confidence, we used the non-repetitive segments from the two ends of these AFGP genes—the 5′ non-repetitive coding regions (signal peptide and propeptide) plus 5′ flanking region near upstream of the start codon (1140 bp), and the proximal 3′ flanking region downstream of the stop codon (325 bp). The 5′ and 3′ set of sequences was aligned separately using MUSCLE v.3.8.31 [20]. Each sequence alignment was then tested for their respective best evolution model of nucleotide substitution using ModelTest-NG v.0.1.6 [21]. Likelihood ratio tests were conducted to compare different models, and the best model was found to be GTR+I+G and HKY respectively for the *AFGP* 5′ and 3′ alignments based on both AICc and BIC. Bayesian and ML phylogenetic analyses were performed separately for each alignment using the same methods as described above for the concatenated *COI* and *ND2* sequences. AFGP gene orthology is then determined by the topology of the constructed gene tree and shared gene features such as the characteristic C-terminus residues.

### 2.7. Codon Usage Analyses of AFGP Tripeptide Repeats

We analyzed general codon usage and frequencies of codon types at each of the three residue positions in the tripeptide repeats in all functional AFGP genes and determined the coding patterns of these tripeptide repeats using the Sequence Manipulation Suite [34].

## 3. Results and Discussion

### 3.1. Gadid Phylogeny and Mapping AFGP Phenotype

We reconstructed the phylogenetic relationships among a broad sampling of gadid species to map the AFGP trait and trace the evolutionary origin and history of gadid *AFGP.* Tree topologies generated by Bayesian, ML, MP, and NJ analyses were essentially congruent and well resolved (Appendix A). The cladogram in Figure 1 represents the summary tree, with node support values given for each of the four analyses. The monophyly of the subfamily Gadinae, and the basal positions of Lotinae and Gaidropsarinae in Gadidae are fully consistent with previous phylogenetic hypotheses of Gadidae [35] and the order Gadiformes [36]. In a recent analysis [37], these subfamilies were elevated to family status, but their phylogenetic relationships remain unchanged. Here we will use the subfamily nomenclature.

The six known AFGP-bearing species (indicated by red +; Figure 1), *Gadus ogac, G. morhua, Arctogadus glacialis, B. saida, Eleginus gracilis,* and *M. tomcod* are all members of the subfamily Gadinae. Except for *G. morhua*, serum samples of these species exhibit characteristic ice crystal growth inhibition (Figure 1, top inset), and substantial levels of thermal hysteresis, at 0.86 °C to 1.1 °C for the Arctic *A. glacialis*, *B. saida, E. gracilis* and the Labrador coast *G. ogac*, and 0.4 °C for the cold temperate *M. tomcod*. AFGPs purified from the same serum samples are of characteristic high yields, giving estimates of circulating blood concentrations of 24.7 mg/mL for *A. glacialis*, 20.5 mg/mL for *B. saida*, 31.1 mg/mL for *E. gracilis*, 22.7 mg/mL for *G. ogac*, and 12.3 mg/mL for *M. tomcod*. *G. morhua* is known to produce AFGPs and its designation of phenotype status (++) were based on thermal hysteresis in the literature [38]. Our serum sample was collected in the summer and had insignificant thermal hysteresis, <0.05 °C. Despite the low thermal hysteresis, a very small amount of AFGPs could be recovered, but they appear as a smear lacking discrete isoforms on a test gel, suggesting seasonal degradation into ineffective remnants. This degradation is consistent with disappearance of serum thermal hysteresis in the summer reported for the Newfoundland *G. morhua* population [38]. Thus, *G. morhua* AFGPs were not included in the gel in Figure 2 as they lack isoform information.

The AFGP isoform compositions revealed by fluorescence gel electrophoresis in this study (Figure 2) are the first documentation and comparison of the full native physiological complement of AFGP size isoforms and their relative abundance in the blood of these five gadids. Each species produces a family of size isoforms akin to the co-electrophoresed Antarctic notothenioid references. Except for *M. tomcod*, the two smallest isoforms, AFGP7 (five tripeptide repeats) and 8 (four tripeptide repeats) are most abundant, similar to Antarctic notothenioids, but the number and abundance of the larger isoforms (AFGPs 6 and AFGPs 1–5 ranges) vary between gadid species. The phylogenetically close *E. gracilis* and *M. tomcod* share quite similar isoform profiles, and both are distinct from the other three gadids in expressing high levels of an isoform in the AFGP 6 range (the most intensely fluorescent band midway down the lane, Figure 2). Serum samples of the majority of the remaining gadid species have insignificant or no measurable thermal hysteresis (+/−), only faceting of single crystal test ice was observed, but ice growth could not be arrested. Two species—*M. proximus* and *L. lota,* were designated (−) (Figure 1), based on no measurable serum thermal hysteresis in one specimen of the former, and assumed absence in the latter. *L. lota* is a (and the only) freshwater gadid. Its equilibrium FP resulting from plasma electrolytes and other osmolytes would be lower that of freshwater (0 °C), not requiring presence of antifreeze proteins.

As the phylogenetic tree shows, the six gadids with strong AFGP phenotype do not form a single clade, but nest with species with little or no antifreeze activity (+/−, or −) and are separated by them in the tree (Figure 1). The *Gadus* clade that contains AFGP-bearing *G. ogac* and *G. morhua* is separated by the AFGP-deficient *Theragra* lineage from the AFGP-bearing high Arctic *Arctogadus/Boreogadus* clade, which in turn is separated from the *Eleginus/Microgadus* clade by the AFGP-deficient *Melanogrammus/Merlangius* and *Pollachius* lineages. In addition, the AFGP phenotype is not consistently present even in closely related species. The robustly AFGP fortified (++++) *E. gracilis* is sister to the AFGP-null (−) *M. proximus,* and the AFGP endowed *G. ogac* is sister to the AFGP-deficient *G. macrocephalus*. On phylogenetic considerations alone, the strong support for sister relationship between *E. gracilis* and *M. proximus* indicates *Eleginus* should be synonymized with *Microgadus*, as has been advocated before [35,36]. In addition, *G. ogac* and *G. macrocephalus* are genetically indistinguishable, with only 3 nt substitutions in the combined ND2 and COI sequences in our analyses, corroborating prior hypothesis that these two are the same species [35,36]. However, their allopatric distribution, disparate body sizes and contrary AFGP phenotypes argue otherwise. Taxonomic resolution aside, our results showing inconsistent pairing of the AFGP phenotype and species phylogenetic grouping suggest two alternate hypotheses of gadid *AFGP* evolutionary history—a single origin and multiple losses, or independent gain in separate gadid lineages, which will be discussed further below.

### 3.2. AFGP Gene Family Genomic Locus

To investigate the *AFGP* evolutionary history in gadids, we sequenced and analyzed the entire *AFGP* genomic loci of three species, *B. saida, G. morhua* and *M. tomcod* representing different subclades (Figure 1). The schematics of their locus organization and gene contents are shown in Figure 3. *B. saida* has 16 AFGP genes (*Bs_AFGP1* to *16*) forming three clusters, with 12 of the genes having intact gene structures and therefore is likely capable of being transcribed. The remaining four are pseudogenes (ψ); *Bs_AFGP6ψ* and *Bs_AFGP14ψ* are both 5′ truncated, while *Bs_AFGP1ψ* and *Bs_AFGP7ψ* have frameshifts due to indels present upstream of the tripeptide coding region. *G. morhua* has seven AFGP genes (*Gm_AFGP1* to *7*), two of which (*Gm_AFGP1ψ* and *Gm_AFGP7ψ*) are pseudogenes and orthologous to *Bs_AFGP1ψ* and *Bs_AFGP7ψ*. *M. tomcod* has four AFGP genes (*Mt_AFGP1* to *4*) and the last one is a pseudogene with a long (6.5 kbp) insertion in the coding region. These results depict a gradation of gene family size and gene copy number, largest in the high Arctic *B. saida*, and smallest in the cold temperate (Shinnecock Bay, NY, USA) *M. tomcod,* correlating with the substantially greater circulatory AFGP we found in *B. saida* (20.5 mg/mL) than in *M. tomcod (*12.3 mg/mL), consistent with the expected differential strength of environmental selective pressures these two species encounter.

### 3.3. AFGP Polypeptide Cleavage Sites and Size Isoforms

To trace the evolutionary processes giving rise to the gadid AFGP gene structures responsible for the observed protein isoforms, we characterized the features of all member genes in the family in the three gadids. In the Antarctic notothenioids, each AFGP gene encodes a large polyprotein precursor consisting of multiple AFGP isoforms linked in tandem by a conserved three-residue linker sequence of predominantly Leu-Ile/Asn-Phe, which are post-translationally cleaved to produce the mature AFGP molecules [12,40]. Like the notothenioid *AFGP*s, the deduced amino acid sequence of gadid AFGP genes in this study shows that the uninterrupted *AFGP* coding region also encodes a large AFGP polyprotein consisting of (Thr-Ala/Pro-Ala)_n_ tripeptide repeats ending with two to four characteristic C-terminus residues, but the linker residues differ (Appendix A). Most of the tripeptide repeats are Thr-Ala-Ala and Thr-Pro-Ala, with an occasional substitution of the Thr by Arg or Lys, leading to the periodic placement of Arg/Lys-Ala-Ala among the (Thr-Ala/Pro-Ala)_n_ repeats. Paired basic amino acid residues—primarily Arg and Lys, are well known to serve as cleavage sites in protein precursors, being recognition sites of trypsin and trypsin-like proteases, which cleave at the carboxyl side of Arg and Lys [41,42]. Previous amino acid composition analyses of purified AFGPs from *B. saida* showed the presence of the three residues Thr, Pro, and Ala of the tripeptide repeats in expected percentages, no detectable Lys, and small amounts of Arg (0.6–1.2%) [43]. Lys and Arg occur at 0.3% and 3.0% respectively in the *AFGP* coding sequences in the full complement of *B. saida AFGP*s analyzed in this study (Appendix A), indicating that the Lys residues and some of the Arg residues are the cleavage sites. We found the codons for Arg and Lys in all AFGP genes in the three gadids were exclusively AGA and AAA, and the predominant codon for Thr is ACA (Appendix A). Thus, a single nucleotide mutation from ACA to AGA or AAA would convert a Thr to Arg or Lys, producing the putative cleavage residues in the polyprotein.

Three different types of cleavage sites are observed in the AFGP genes—single Arg, single Lys, and Arg-Ala-Ala-Arg. The single Arg sites occur in all *B. saida* and *G. morhua* AFGP genes except *Bs_AFGP5* and *Gm_AFGP5*; the single Lys sites are present in all genes in *B. saida AFGP* cluster I, all *G. morhua* genes and the first two genes in *M. tomcod*; and the Arg-Ala-Ala-Arg sites only appear in the genes in *B. saida* cluster II and III, and *Mt_AFGP3* (Appendix A). If all the potential cleavage sites are cleaved after translation, *B. saida* can produce 119 peptide molecules with tripeptide repeats ranging from four to 33 to be glycosylated and become mature AFGPs. Four (Thr-Ala/Pro-Ala) repeats correspond to the smallest isoform AFGP8 (~2600 Da), and the largest isoform with 33 repeats would have a molecular mass of ~20,000 Da, consistent with the range of size isoform obtained from purified serum AFGPs in this and our prior study (Figure 2) [43]. *G. morhua* AFGP genes can lead to 23 AFGP molecules with tripeptide repeats ranging from four to 45, which would span a size isoform profile comparable to the congeneric *G. ogac* (Figure 2).

The case of *M. tomcod* appears peculiar. It has two genes encoding 53 and over 200 tripeptide repeats respectively, each with only one single Lys near the N-terminus, which could lead to a mature glycosylated protein of ~32,000 Da and >121,000 Da respectively. A 32,000 Da isoform would be consistent with the high end of the AFGP1–5 range in the *M. tomcod* AFGP protein profile (Figure 2). However, none of the bands in the AFGP protein profile would match a 121,000 Da isoform. Thus, it is currently unclear how this putative large isoform is processed if it were expressed. A third gene encodes 65 small AFGP molecules with tripeptide repeats ranging from four (AFGP8) to nine (~5400 Da, in the AFGP6 range), if all cleavage sites are cleaved. The 9-repeat isoform would correspond to the most abundant AFGP6 in the gel (Figure 2). However, the AFGP profile also shows presence of many isoforms larger than this AFGP6, suggesting not all putative cleavage sites are cleaved. Exactly how this can be achieved to produce the larger isoforms from the same polyprotein precursors is again currently unknown. The agreement between predicted AFGP isoforms in encoding genes and the isoform profile of the purified proteins shared by *B. saida* and *G. morhua* indicate these two species likely derived their AFGP genotype from their common ancestor at node B (Figure 1). The distinctive features of two of the three *M. tomcod* AFGP genes, and the peculiar discordance between predicted AFGP isoforms and mature protein profile suggest the AFGP genotype might have separately developed in the ancestor of the *Eleginus/Microgadus* clade at node C (Figure 1). Corroborating evidence for this hypothesis could be gleaned from future characterization of the AFGP genes and genomic locus of *E. gracilis.*

The number of encoded tripeptide repeats varies substantially among paralogs in each species, and between orthologs in *B. saida* and *G. morhua*. This indicates that the tripeptide repetitive regions likely expanded independently in individual genes by internal duplications within the polypeptide coding region, and that intragenic and whole gene duplication likely continued in a species-specific manner after speciation.

### 3.4. Codon Usage of AFGP Tripeptide Repeats

To investigate the evolution of AFGP tripeptide repeats, we examined codon usage frequencies of each amino acid in the tripeptide units as well as and tripeptide coding unit frequencies for all the AFGP genes in the three gadids. While the encoded tripeptide repeats are conserved as either Thr-Ala-Ala or Thr-Pro-Ala, the nine nucleotides that code for each tripeptide unit vary, and therefore the nucleotide sequences of the *AFGP* coding region usually consist of imperfect tandem repeats.

Codon analyses of the three amino acid positions of tripeptides for all AFGP genes from the three gadids reveal strong codon bias (Appendix A). First, some common biases were observed in all genes, suggesting mutual evolutionary constraint of codon usage existed in these species. For instance, most AFGP genes in the three species have neither codon ACG for Thr the first residue, nor CCC for Pro the second residue. A strong bias against the codon GCT for Ala and CCT for Pro in the second position was also observed (orange shades in Appendix A). Second, species-specific biases were observed in *M. tomcod* (blue shades in Appendix A), suggesting independent gene evolution in different clades. Third, similar codon usage patterns were observed between the orthologous gene pairs in *B. saida* and *G. morhua* (indicated as same non-black colors in Appendix A), and among closely related paralogs that have been minted recently via duplications, such as the genes in *B. saida AFGP* cluster II and III (Bs_AFGP8 to Bs_AFGP16), and Mt_AFGP1 and Mt_AFGP2. These similar patterns of codon usage corroborate the relationships we observed in AFGP gene phylogeny (Figure 4). Lastly, outlier genes with distinct codon pattern are observed in each species, such as Bs_AFGP5, Gm_AFGP6, Mt_AFGP3, suggesting that individual genes have experienced internal tripeptide repeat expansion independently. In sum, different patterns of duplication and mutation occurred throughout the evolutionary process of the AFGP gene family in the gadid lineage, leading to the high molecular heterogeneity of the AFGP tripeptide coding sequence (CDS).

**Figure 4 genes-12-01777-f004:**
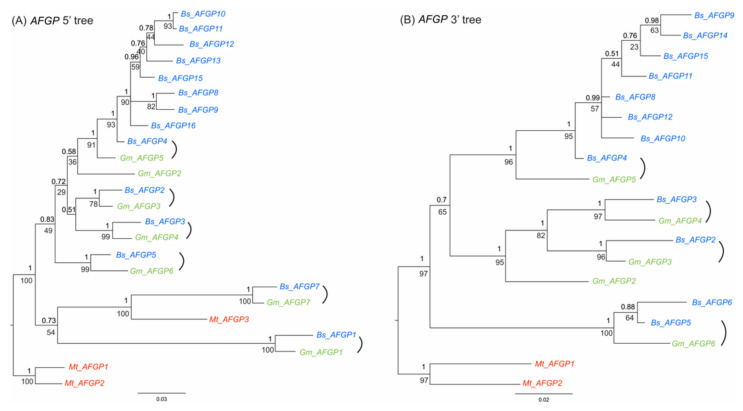
Phylogenetic trees of AFGP gene families in three gadids. Congruent tree topology was inferred from Maximum likelihood and Bayesian analyses. The Bayesian posterior probability values (top) and Maximum likelihood bootstrap values (bottom) are labeled at branch nodes. Branch length information from the Bayesian tree is shown. AFGP genes in *B. saida, G. morhua,* and *M. tomcod* are shown by blue, green, and red, respectively. Brackets indicate the orthologous gene pairs of *B. saida* and *G. morhua*. (**A**) AFGP 5′ tree was inferred using the nucleotides sequences of 5′ non-repetitive region (~450 bp) from exon 1 to the start of tripeptide repeat coding region in exon 3, encompassing the 5′ untranslated region (UTR), signal peptide sequence, and the propeptide. *Bs_AFGP6ψ*, *Bs_AFGP14ψ* and *Mt_AFGP4ψ* are 5′ truncated pseudogenes and therefore were not included in the 5′ tree. (**B**) AFGP 3′ tree was inferred using the nucleotides sequences of 3′ flanking region (~325 bp) from the stop codon to the region where the sequence similarity ends, including 3′ UTR and downstream sequence. *Bs_AFGP1ψ*, *Bs_AFGP7ψ*, *Gm_AFGP1ψ*, *Gm_AFGP7ψ*, *Mt_AFGP3* and *Mt_AFGP4ψ* were excluded from the *AFGP* 3′ tree construction because their 3′ flanking regions share no sequence similarity with the other AFGP genes. The assembly of *Bs_AFGP16* is not complete, missing the 3′ portion, and thus it is not included in the 3′ tree.

### 3.5. Molecular Mechanisms of the Duplication and Stabilization of the Tripeptide Repeats

Given that the tripeptide repeats are the functionally essential sequences in the new gene, we also deduced the possible molecular mechanisms of their formation. Slipped-strand mispairing events, in concert with unequal crossing-over, can readily account for rapid expansion of the tripeptide coding repeats. Tandem repeat tracts containing motifs that differ by a few substitutions such as the imperfect 9-nt repeats of AFGP tripeptide CDS could be the consequence of multiple slipped-strand mispairing events occurring before and after base-substitution events. A substitution can create new repeat units from existing ones, and subsequent slippage events, which repetitive regions are prone to, can then expand these new motifs, resulting in a new tandem repeat adjacent to the old one. Slipped-strand mispairing during DNA replication is considered the likely major process in the initial expansion of short repeats, and further expansion thereafter likely results from unequal crossing over, which simple tandem repeats are predisposed to [44]. Although the shared tree topology between *AFGP* 5′ and 3′ sequences (Figure 4) suggests tripeptide CDS expansion by non-allelic unequal crossing-over to be unlikely, we cannot rule out the possibility of unequal crossing-over between different portions of tripeptide repeats in the allelic genes.

Long stretches of direct repeats are generally unstable as highly repetitive regions are prone to rearrangements by homologous recombination. If the rearrangement occurs in the coding region, reading frame shifts could occur resulting in loss of gene function. However, no gene rearrangement was observed in the *AFGP* loci, which suggests the presence of selective constraint on coding region recombination. The interspersed Arg-Ala-Ala or Lys-Ala-Ala among the Thr-Ala-Ala repeats and Thr-Pro-Ala replacements of Thr-Ala-Ala could all increase the stability of tandem repeats in AFGP genes because they reduce sequence similarity between duplicons. Support for this model has been established in many instances of human disease alleles, where the loss of interspersed interruptions in sequence repeats significantly increase the instability of the repetitive sequence resulting in disease condition [45,46]. Besides the imperfect sequence repeats, distinct tripeptide repeat numbers in each AFGP polyprotein coding region may also increase the stability of the AFGP gene family, preventing deleterious deletions or non-functionalizing rearrangements. The rapid divergence of tripeptide repeat CDS in AFGP genes has furthermore led to a situation, where the major coding block has evolved even faster than the non-coding 5′ and 3′ UTR of the gene. On all accounts, the variation in the number of tripeptide repeats leads to length isoforms, and the variation in residue composition of repeat patterns leads to compositional isoforms. Both contribute to a high degree of *AFGP* heterogeneity despite the simplicity of the tripeptide monomer, and in turn enhance the evolutionary stability of the *AFGP* family.

### 3.6. Expansion of the AFGP Gene Family and Genomic Locus

To deduce how the gene family expanded under selection, we analyzed the large *AFGP* genomic locus of the high-Arctic species *B. saida.* High protein levels of AFGP were achieved by expansion of the AFGP polyprotein coding region through intragenic duplications, as well as by increasing gene copy number through whole gene duplications. The reconstruction of the *B. saida AFGP* genomic locus produced three unconnected AFGP gene clusters in three sequence contigs, which may be separated by long distances of sequences without AFGP CDS. The evolutionary relationships of the 16 AFGP genes and pseudogenes in *B. saida* were analyzed using the 5′ (from UTR to the start of tripeptide repeats) and 3′ flanking regions, which share sequence homology in all the genes (Figure 4). We deduced a plausible process of AFGP gene family expansion in *B. saida* (Figure 5) based on the *AFGP* 5′ and 3′ tree topology (Figure 4) and additional information from further upstream and downstream flanking regions sharing longer sequence similarity between some gene members. The two pseudogenes *Bs_AFGP1ψ* and *Bs_AFGP7ψ* occupying the two ends of cluster I, are potentially the most basal copies among all *AFGPs*. Lacking selective restriction on these basal pseudogenes allows accumulation of multiple nucleotide substitutions, reflected in their long branch lengths in the tree. These two pseudogenes have relatively shorter and more variable tripeptide repeats and share a similar insertion in the 5′ non-repetitive region. Their ancestral paralog likely gave birth to the first functional *AFGP*, which then proliferated throughout the genomic region between *Bs_AFGP1ψ* and *Bs_AFGP7ψ*, giving rise to the other five genes (*AFGP 2–6*) in cluster I. The *AFGPs* in clusters II (*Bs_AFGP8* and *9*) and III (*Bs_AFGP10* to *16*) were likely derived from *Bs_AFGP4* in cluster I, as it is recovered as the closest sister gene to the two clusters in both trees (Figure 4). Among clusters II and III *AFGP* copies, *Bs_AFGP16* shares the longest 3′ flanking sequence homology with Bs_AFGP4 and thus is likely the immediate duplicon from *Bs_AFGP4*. Further duplications of *Bs_AFGP16* produced the other six genes (*Bs_AFGP10* to *Bs_AFGP15*) in cluster III. The gene expansion in cluster III likely occurred quite rapidly as all these genes share highly conserved flanking sequences and therefore the relationships among these genes cannot be well resolved by the phylogenetic analysis (Figure 4). 

The formation of cluster II (*Bs_AFGP8* and *9*), which spans 7.5 kbp, is likely through duplication and translocation of a 7.5 kbp region containing one pair of adjacent genes similar to the three extant pairs *Bs_AFGP10* and *11*, *Bs_AFGP12* and *13*, and *Bs_AFGP15* and 16 in cluster III. This is supported by very high sequence similarity shared by *Bs_AFGP8* and *9* with the cluster III gene pairs at a 1.4 kbp intergenic region and 1.3 kbp sequences immediately upstream and downstream of the gene pairs. The 7.5 kbp *AFGP* cluster II is flanked by low-complexity sequences—a long stretch of mononucleotide repeats (A)_n_ and dinucleotide repeats (GT)_n_ at each end. Poly-purine (or poly-pyrimidine) and alternative purine-pyrimidine tracts are known for potentially generating breaks in DNA because they can form non-B-form DNA conformations [47]. Thus, the 7.5 kbp *B. saida AFGP* cluster II might have been inserted at a double-strand DNA breakage site in these low-complexity sequence regions.

Through comparing the intergenic sequences between *AFGPs* within each cluster and among different clusters in *B. saida*, we found greater frequencies of nucleotide substitutions, sequence insertions and deletions in the basal cluster I than in the derived clusters II and III, suggesting these sequence mutations have been accumulated continuously. Insertions and deletions in the intergenic sequences also contribute to an increase in the variation between duplicons and thus decreased the chance of gene loss due to homologous recombination [48]. Together with the individual intragenic tripeptide variations, these intergenic sequence variations appeared to be also selected upon for preserving structural integrity of the AFGP genes.

### 3.7. AFGP Gene Evolutionary History in Gadid Lineage

The role of the environmental selective pressure on the expansion of *AFGP* family in gadids became clear when we compared the *AFGP* family in the high-Arctic species *B. saida* with the north cold temperate species *M. tomcod* and the geographically widely distributed species *G. morhua*. Among the three gadids, *B. saida* has the largest *AFGP* locus, comprising 16 *AFGP*s in three spatially separate clusters with a combined length of over 500 kbp. The *AFGP* locus in *G. morhua* contains seven *AFGP*s within a 100 kbp distance. *M. tomcod* has the smallest *AFGP* locus, comprising only four *AFGP*s within 30 kbp. The remarkably different *AFGP* copy numbers in these gadids appear to be commensurate with the expected differential selective pressures each lineage experiences. The high-Arctic *B. saida* clearly had undergone the most extensive gene duplication. It is a cryopelagic species, prevalent in the coldest waters associated with the surface pack ice in high Arctic seas [49,50]. The strong selective pressure from the most severe polar condition and the pressing demand for an abundance of the protective protein likely contributed to the expansive *AFGP* locus and the large number of *AFGP*s in *B. saida.* In contrast, *G. morhua* has discrete populations and occurs throughout a wide variety of habitats, from the North Atlantic Ocean to the Arctic area. The individual whose *AFGP* locus we characterized is from a stock called Norwegian coastal cod (NCC) [51]. NCC are generally found in fjords and along the coast of Norway [52] with only limited sea-ice coverage during winter and spring [53]. Thus, the less severe habitat condition is consistent with the smaller *AFGP* family we found in this species. Among the three species, *M. tomcod* inhabits the lowest latitudes with mild water temperatures along the coastal areas in the western Atlantic, experiencing only occasional freezing conditions in shallow waters in the winter. This least severe environment would result in less protective AFGP demand, correlating with the smallest *AFGP* family we found in *M. tomcod.* This direct relationship between the *AFGP* family size, protein product, and the severity of environmental freezing conditions is illustrative of a clear linkage between adaptive functional evolution and natural selection pressures.

The *AFGP* loci of the *G. morhua* and *M. tomcod*, and *B. saida AFGP* cluster I share high sequence identity (85–99%), indicative of homology. Microsynteny of this genomic region is further corroborated by shared neighboring hypothetical protein-coding genes, including *MAK16* and *RAB14* upstream from *AFGP* locus and the downstream actin filament-associated protein we have reported [19]. Thus, the *AFGP* in the three species likely evolved from the same genomic site, supporting the hypothesis of a single ancestral origin of *AFGP* in gadids. However, results from the deep analyses in this study indicate the subsequent gene family propagation and locus expansion process occurred in distinct paths in different gadid phylogenetic clades. Six of the seven *AFGPs* in *B. saida AFGP* cluster I have orthologs in *G. morhua* (Figure 4), suggesting these genes have evolved in the common ancestor of these two species. Strikingly, none of the *B. saida/G. morhua* orthologs are found in *M. tomcod*, indicating that this species had undergone a different mode of *AFGP* locus expansion. The *AFGP* orthologs of *M. tomcod* are found in another AFGP-bearing gadid *E. gracilis*, whose AFGP genes were sequenced in our previous study [29,54]. *B. saida* and *G. morhua* are more closely related, belonging to the same clade (clade B, Figure 1) with AFGP-bearing species, while *M. tomcod* and *E. gracilis* are closely species in the *Microgadus/Eleginus* clade (clade C, Figure 1). Thus, we can deduce that the *AFGP* family has propagated independently in the species from the two AFGP-bearing gadine clades, although they share a single incipient ancestor (at node A, Figure 1).

Finally, the *AFGP* family in *B. saida* further expanded to form two additional new clusters—clusters II and III, which have no orthologs in *G. morhua* or *M. tomcod.* We searched the *G. morhua* genome [51,55] for similar sequences to *B. saida* clusters II and III, and found the *G. morhua* orthologous region corresponding to the flanking regions of *B. saida AFGP* cluster II but it contains no *AFGP*s. The two *AFGP*s in cluster II likely formed through insertion in an originally *AFGP*-free genomic region of *B. saida*. This new insertion (cluster II) as well as the formation of cluster III in *B. saida* likely occurred and became fixed in the population when the *Boreogadus* lineage continued to encounter severe freezing selection pressure after speciation of the respective ancestor of *B. saida* and *G. morhua.*

## 4. Conclusions

In summary, the evolutionary history of the de novo AFGP gene in the gadid lineage is complex and dynamic, as the results of the in-depth analyses in this study indicate. Under selective pressure from marine glaciation, the primordial *AFGP* originated de novo once, in a common gadine ancestor (Figure 1 node A), followed by retention or degeneration of the new trait in different lineages. The specific evolutionary trajectory of the new gene hinged on the changing local selective pressures. It degenerated in lineages whose habitats became non-freezing during subsequent interglacial periods, or in those that moved to non-freezing habitats, where maintenance of the AFGP trait is unnecessary and costly. In others that remained in icy environments such as the high Arctic species, or would encounter winter freezing conditions in cold temperature latitudes such as *M. tomcod*, the selective force on the newly minted gene persisted, driving its propagation into a gene family and different polypeptide isoforms through gene duplication and intragenic tripeptide expansion. These evolutionary events are clearly illustrated by the two subclades with extant AFGP-bearing species nesting with AFGP- deficient species (Figure 1, node B and node C subclades). Some species in these two subclades lost their AFGP genotype and phenotype, while the trait expanded in the *Arctogadus/Boreogadus/Gadus* lineages and *Eleginus/Microgadus* lineages and persisted to the present and is commensurate with the severity of their respective habitats. The complex dynamics of glacial advances and retreats in the northern hemisphere throughout the Quaternary epoch likely produced temporally and spatially separate freezing selective pressures on closely related species, leading to the distinct modes of *AFGP* family expansion in these different gadid subclades we see today.

## Figures and Tables

**Figure 1 genes-12-01777-f001:**
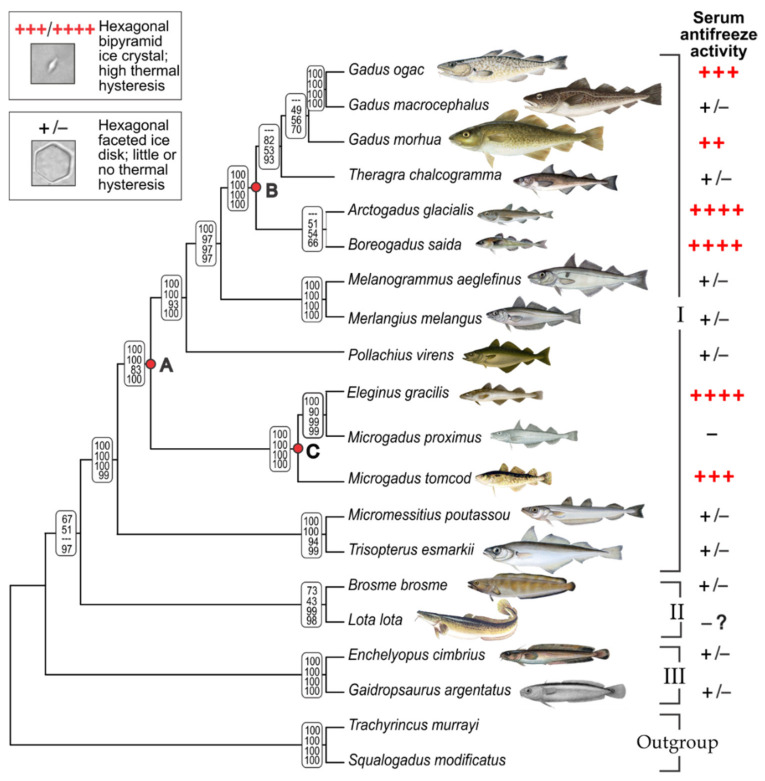
AFGP phenotype on gadid phylogeny framework. The cladogram summarizes congruent topologies of four phylogenetic trees in Appendix A. The three gadid subfamilies are indicated: I, Gadinae; II, Lotinae; III, Gaidropsarinae. Nodes marked A, B, C represent AFGP-bearing ancestors (details in text). Gradation of AFGP phenotype mapped to species as follows: +++ and ++++ indicate high levels of serum antifreeze activity measured as thermal hysteresis, strong growth inhibition of test ice crystal leading to a minute hexagonal bipyramid (top inset, illustrating *B. saida*), and the full complement of discrete AFGP isoforms can be purified from serum (see Figure 1B); ++ designation of *G. morhua* was based on published thermal hysteresis in the literature [38], because our samples were from summer specimens when antifreeze activity has declined (see text); +/− indicates little or no thermal hysteresis, hexagonal faceting of test ice crystal is induced (lower inset, illustrating *M. aeglefinus*) in serum of some individuals but ice expansion cannot be arrested. Minus sign indicates no ice faceting, i.e., no antifreeze activity. Absence of ice activity in *Microgadus proximus* was based on a single sample. No serum sample was available for the freshwater species *Lota lota.* Simce antifreeze protection would not be required in freshwater habitats, its activity designation was indicated with (− ?) since it was assumed negative.

**Figure 2 genes-12-01777-f002:**
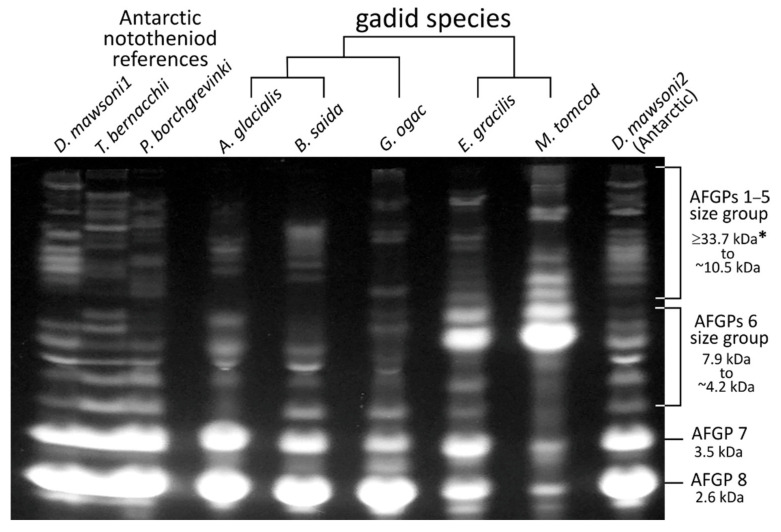
Gadid AFGP isoform composition and abundance. Gradient gel electrophoretogram of ~300–400 g of purified and fluorescently labeled AFGPs from blood serum of five gadid species (+++/++++ phenotype), with evolutionary relationship depicted by the cladogram. AFGPs do not migrate on typical SDS protein gel, thus previously characterized AFGPs of three high-Antarctic notothenioid fishes (*Dissostichus mawsoni, Trematomus bernacchii and Pagothenia* (reclassified as *Trematomus*) *borchgrevinki*)) were co-electrophoresed as references for isoform and isform group delineation. * Nomenclature of AFGP isoform numbers and isoform grouping is based on historical designation for the Antarctic notothenioid *P. borchgrevinki* AFGPs, for which the MW of some isoforms had been determined by ultracentrifugation sedimentation methods [10,39]: AFGP7 (five tripeptide repeats) and AFGP8 (four tripeptide repeats) are the smallest isoforms at ~3.5 kDa and ~2.6 kDa respectively, AFGP6 isoform group spans 4.2–7.9 kDa and the large AFGP 1–5 group spans 10.5 to ≥33.7 kDa. The AFGP isoforms in all species shown comprise the physiological mixture in blood circulation, with relative abundance of each isoform (band) within the species being proportional to its fluorescence intensity on the gel. In the Antarctic species, the two smallest size isoforms—AFGP 7 and AFGP 8 predominate, and the larger sizes of AFGPs 1–6 occur in lower levels. The gadids, except for *M. tomcod*, also synthesize abundant AFGP 7 and 8, but the number and abundance of the larger isoforms vary between species. The phylogenetically close *A. glacialis* and *B. saida*, and *E. gracilis* and *M. tomcod* share comparable isoform profile. *E. gracilis* and *M. tomcod* are distinct from the other three gadids in expressing particularly high levels of one AFGP 6 isoform (the most intensely bright band in the AFGP 6 size group), estimated to consist of nine tripeptide repeats (five more than AFGP 8; the fully resolved AFGPs in the AFGP 6–8 range are an approximate ladder of one tripeptide repeat increment).

**Figure 3 genes-12-01777-f003:**
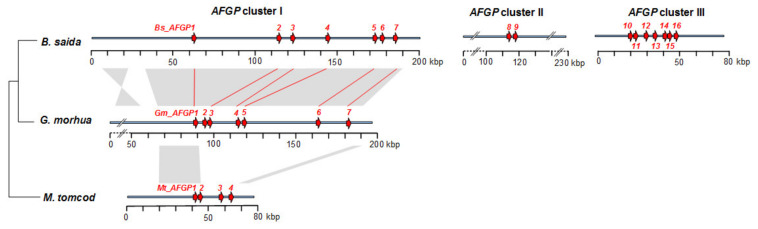
AFGP gene family expansion history in gadid lineage. AFGP genes are represented by the red arrows. Orthologous AFGP genes deduced from AFGP gene phylogenetic analysis (Figure 4) are connected by red lines. Grey shaded areas indicate regions of high nucleotide identities (85–99%) between the *AFGP* locus of *G. morhua* and *M. tomcod*, and *B. saida AFGP* cluster I. The phylogenetic relationship of the three species is indicated by the lines on the left of the species names.

**Figure 5 genes-12-01777-f005:**
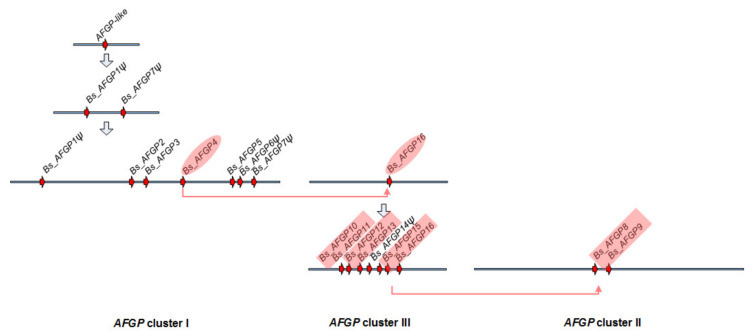
Plausible process of AFGP gene family expansion in *B. saida.* Grey arrows from top to bottom suggest the order of hypothetical status during AFGP gene family expansion. The three AFGP gene clusters are shown from left to right. AFGP genes are represented by the red arrows. Genes with a ‘ψ’ symbol indicate they are pseudogene. Genes in pink shades connected by an arrow share the most similar gene features and high sequence similarity.

## Data Availability

The obtained gadid COI gene sequences were deposited to GenBank under the accession numbers MK011273–MK011290.

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
