# Peer review of "Propagation of a De Novo Gene under Natural Selection: Antifreeze Glycoprotein Genes and Their Evolutionary History in Codfishes"

_genes, 2021, doi:10.3390/genes12111777_

Round 1
Reviewer 1 Report
In this thoughtful and highly original paper Zhuang and Cheng provide a plausible scenario for the evolution of the AFGP gene family in gadids. Based on extensive analyses they hypothesise that the primordial AFGP gene originated once de novo, and then evolved within different lineages reflecting local selective pressures (essentially the extent of exposure to freezing conditions).
As noted by the authors, the possibility exists that the AFGP genotype might have evolved separately in the ancestor of the Eleginus / Microgadus clade. They suggest further work to resolve this possibility, which leaves a degree of uncertainty but does not detract from the main thrust of the manuscript.
Given that the AFGPs exist in all species studied to date as a family of size isoforms composed of a tripeptide (Thr-Ala/Prol-Ala) repeat, attention is devoted to the possible molecular mechanisms underlying tripeptide expansion. The suggested pathway via slipped-strand mispairing events combined with unequal crossing-over seems entirely plausible. It is proposed that the tripeptide then expanded independently in individual genes through duplication events and that intragenic and whole gene duplication continued in a species-specific manner. Examination of codon frequencies for each amino acid in the tripeptide sequence of the three gadids studied suggested that patterns of duplication and mutation in their AFGP family continued during their evolution, thus accounting for the observed heterogeneity. Analysis of the AFGP gene locus in the high-Arctic B. saida provided an understanding of how the gene family expanded, and comparison with the AFGp gene family in the cold temperate M. tomcod and the relatively widely-distributed G. morhua offered a way of understanding the role of environmental selective pressure on AFGP gene family expansion. From their analysis, the authors provide further support for a single ancestral origin of the AFGP gene in gadids, followed by independent propagation dependent on local selective pressures, resulting in degradation (non-freezing habitats) or expansion (freezing habitats).
The AFGPs are synthesised as a polyprotein precursor, which is then processed into different-sized isoforms through cleavage at specific linker sites (Arg; Lys; Arg-Ala-Ala-Arg). Sequence analysis showed that the codon for Thr (predominantly ACA) could yield AGA (coding for Arg) or AAA (Lys) by a single nucleotide mutation, thus yielding putative linker cleavage sites. Interestingly, one polyprotein in M. tomcod would yield a mature AFGP >121 kDa, which is outwith the current known AFGP size range. This begs the question as to how it is processed, for which the fallback position is that it may not be expressed, although this too leaves uncertainties. Additionally, electrophoretograms show an inexplicable range of larger isoforms suggesting that not all cleavage sites are actively processed. How this could be achieved is uncertain. Again, I do not consider these uncertainties to be significantly limiting: to me they encourage future researchers in the field and we should focus on the many positives in this manuscript.
Minor suggestions
Ln10 The de novo birth
Ln10 DNA is an important
Ln11 However,
Ln33 emergence of a de novo gene
Ln39 lack of an a priori clue
Ln44 natural selection to act.
Ln47 fishes – was
Ln47 [9], and was followed
Ln57 freezing points (FPs)
Ln60 colligative FPs
Ln68 under a clear
Ln71 breadth
Ln76 However
Ln96 the ancestral Antarctic
Ln100 “Inventive” is probably not the right word here. A neutral word, not implying the existence of an inventor, would be “interesting”.
Ln112 a broad sample of codfishes
Ln126 using an appropriate
Ln147 and were read
Ln275 The assumption with regard to L. lota is noted. If samples were not available this should be made clear.
Ln289 The phrasing of this sentence is confusing for the reader as it implies that G. morhua serum does not exhibit ice crystal growth. The authors should explain here (not later) the reason for this particular result (i.e., available G. morhua samples were collected in the summer).
Ln299 Pagothenia (Trematomus) borchgrevinki might be more accurate for future readers.
Ln324 seasonal degradation or loss (e.g via excretion) or both? Aren’t AFGPs relatively resistant to degradation?
Ln325 There is some lack of clarity here as +/- is described by the authors as having “little or no hysteresis”. Here, it is taken to mean no hysteresis. Could I suggest “lack significant hysteresis” or perhaps write in terms of faceting.
Ln335 Is G. macrocephalus actually lacking AFGPs, as it scores +/-? Or are the levels minimal?
Ln335 On phylogenetic consideration
Ln460 tripeptide coding sequence (CDS)
Ln478 AFGPs essentially have two parts, the repeating tripeptide and the disaccharide on the Thr. At some point, perhaps not here, an explanation for the existence of a particular disaccharide (galactosyl-N-acetylgalactosamine) might be forthcoming.
Reviewer 2 Report
This is a very interesting paper about the evolution of AFGP genes in codfish. It describes a tremendous amount of genetic, genomic, and protein chemistry work as well as subsequent analyses (phylogenetics), and would represent a substantial addition to the literature. The methods are appropriate and described in sufficient detail, with minor exceptions mentioned below. I have two overall concerns and a number of smaller ones that I believe would improve the paper.
Overall concerns:
- Overall, this is a dense, complex read and would benefit from some economization. I have made a couple of suggestions below about segments that do not seem to add much to the paper.
- The paper would benefit from a little more background on the non-AFGP-expressing species in the major AFGP-expressing clades (e.g., macrocephalus). Are they all non-arctic species that are likely to have lost the genes? Were they tested for AFGP activity in this study or elsewhere? Have their genomes been sequenced or tested for presence of AFGP genes, here or elsewhere? Some of these species are mentioned briefly in the paper, but without significant context.
Minor comments, by line number:
33-34: …”previously deemed implausible”. I was skeptical about this, but it is an accurate interpretation of statements by Monod in the reference. Still, “implausible” means more than “unlikely” and reflects a particular viewpoint that is not necessarily universal. It might be safer to say that de novo origination was previously considered a rare event and now appears to be more common than previously appreciated.
71: “breadth” is likely the intended word, rather than “breath”
73-79: The examples of type II and III AFPs do not appear to be directly relevant. I would suggest removing this paragraph and inserting the lead sentence elsewhere, if needed.
184: Which species were tested?
213-217: Some details that would be required for repetition are missing. This sounds like standard denaturing SDS-PAGE, but that is not mentioned. That detail would be helpful, as would gel & buffer suppliers, loading buffer, and denaturing conditions.
246-247: “The highly repetitive”… is redundant with 242-243 and can probably be eliminated.
268-272: There is no explanation of what ++ (two plus signs) means, or how antifreeze activity can be quantified when none was demonstrated in the study. The authors explain later why they may not have been able to detect antifreeze activity in this species, but it leaves a conceptual gap in the figure legend.
273-275: Plus/minus sign or minus sign is result of testing in this study or in the literature?
287-289: These are all AFGP-bearing species. Were the “non-AFGP” species also tested? I use quotes here because very little information is given about those species or about how the authors know that they do not have AFGP genes or possess AFGP activity.
Figure 2: If this is SDS-PAGE, why is there no size ladder? If not, is there another appropriate standard?
321-322: The authors explain why G. morhua does not demonstrate antifreeze activity in this study, but not why they decided to represent it with two plus signs (see next comment).
Figure 3: It is not clear why the x axis for AFGP cluster II is so wide, when this cluster contains only two adjacent genes. This could be reduced, which would allow expansion of the portion of the figure depicting AFGP cluster I. This cluster is much more complex and interesting, and it would be nice to be able to see it better. The labels within the figure are also quite small.
409-413: This section contains a number of minor issues, including singular/plural usage. Main concern is in line 413. I believe the authors are saying that the gene with 200-tripeptide repeats is inconsistent with the observed protein isoforms, but the wording is unclear.
479-492: This paragraph contains a great deal of speculation about an issue that is indirectly and unresolvedly addressed in the manuscript. I believe it could be removed to improve focus on the paper’s key issues.
489: AFGP unnecessarily mentioned twice in the same sentence.
602: This would be another place to mention what happened in the AFGP-free species, if they are known to be so.
Reviewer 3 Report
In this paper, Zhuang and Cheng examine some aspects of the evolutionary history of the de novo AFGP gene family. They nicely demonstrate the AFGP phenotype of gadids in freezing and non-freezing habitats. They then examine the AFGP gene family members more extensively in three species. I think this will be a fine contribution to genes, but I believe the manuscript requires the following few minor changes / additional clarification.
1) Section 2.7 on AFGP codon usage (section 2.7 beginning on line 266) is incomplete. Maybe it is a remnant of something meant to be deleted? Section 3.4 has a similar title.
2) Figure 2 should contain kDa markers and additional labels describing what I am assuming are different isoforms for AFGPs 1-5 and 6 (?). Could you describe a little bit more about what all the larger-weight bands for lanes for AFGPs 1-5 and 6 are for naive readers? I suggest re-organizing this figure such that a phylogeny can be placed above with the species names. Having a simple table of AFGP concentrations in the blood underneath the figure with each species would also be more helpful to readers.
3) Similar to point 2 above. Line 344: what is “2-3 AFGP 6 isoforms?” I’m assuming these are the names of specific isoforms, but can you label them on the gel in Figure 2? It’s unclear where AFGP 5 isoforms technically “end” and where AFGP 6 isoforms “begin” as they are so close in size for some species if I am interpreting this correctly.
4) Line 381: Some might argue that an intact gene structure does not necessarily mean a gene is functional. in this case, it is more accurate to say that it is likely capable of being transcribed
5) Figure 3 is repeated twice. There’s a smaller version of the figure on top and a larger one below. I assume this was by accident.
6) In Line 388: here I would suggest to say gene copy number rather than gene dosage. Although B. saida does produce larger total amounts of AFGP from circulating blood concentration, it hasn’t been proven whether this is the result from increased protein production from each gene copy. In any case, it might be good to remind readers here that there is an obvious gradient in overall AFGP in blood too
7) for the phylogeny in figure 1, it might be nice to have a description of habitat ( for ex. which clades are in freezing vs. non-freezing habitats--maybe you can change the text of the species to certain colors to depict this).
8) Figure 4 is cut off at the top and the resolution is low.
